# Occurrence of Cortical Arousal at Recovery from Respiratory Disturbances during Deep Propofol Sedation

**DOI:** 10.3390/ijerph16183482

**Published:** 2019-09-18

**Authors:** Ryuma Urahama, Masaya Uesato, Mizuho Aikawa, Reiko Kunii, Shiroh Isono, Hisahiro Matsubara

**Affiliations:** 1Department of Frontier Surgery, Graduate School of Medicine, Chiba University, 1-8-1 Inohana-cho, Chuo-ku, Chiba 260-8670, Japan; urahamaryuma@gmail.com (R.U.); uesato@faculty.chiba-u.jp (M.U.); matsuhm@faculty.chiba-u.jp (H.M.); 2Healthcare Center, Japan Community Healthcare Organization Chiba Hospital, 682 Nitona-cho, Chuo-ku, Chiba 260-8710, Japan; may.water514@gmail.com; 3Clinical Laboratory, Seirei Sakura Citizen Hospital, 2-36-2 Eharadai, Sakura 285-8765, Japan; rk1024ok@gmail.com; 4Department of Anesthesiology, Graduate School of Medicine, Chiba University, 1-8-1 Inohana-cho, Chuo-ku, Chiba 260-8670, Japan

**Keywords:** sleep-disordered breathing, obstructive sleep apnea, arousal, sedation, propofol

## Abstract

Recent evidences suggest that non-arousal mechanisms can restore and stabilize breathing in sleeping patients with obstructive sleep apnea. This possibility can be examined under deep sedation which increases the cortical arousal threshold. We examined incidences of cortical arousal at termination of apneas and hypopneas in elderly patients receiving propofol sedation which increases the cortical arousal threshold. Ten elderly patients undergoing advanced endoscopic procedures under propofol-sedation were recruited. Standard polysomnographic measurements were performed to assess nature of breathing, consciousness, and occurrence of arousal at recovery from apneas and hypopneas. A total of 245 periodic apneas and hypopneas were identified during propofol-induced sleep state. Cortical arousal only occurred in 55 apneas and hypopneas (22.5%), and apneas and hypopneas without arousal and desaturation were most commonly observed (65.7%) regardless of the types of disordered breathing. Chi-square test indicated that incidence of no cortical arousal was significantly associated with occurrence of no desaturation. Higher dose of propofol was associated with a higher apnea hypopnea index (*r* = 0.673, *p* = 0.033). In conclusion, even under deep propofol sedation, apneas and hypopneas can be terminated without cortical arousal. However, extensive suppression of the arousal threshold can lead to critical hypoxemia suggesting careful respiratory monitoring.

## 1. Introduction

Patency of the pharyngeal airway is state-dependent in patients with obstructive sleep apnea (OSA). Reduction of upper airway (UA) dilating muscle activity during loss of consciousness appears to account for UA obstruction during sleep [1]. Burst of UA muscle activity upon cortical arousal has been believed to be an important mechanism for recovery from the pharyngeal obstruction at least in sleeping adult OSA patients while young OSA children resolve obstructive breathing without cortical arousal [2,3]. 

Younes, however, recently proposed that cortical arousals are coincidental events that occur just prior to arousal-independent airway opening through analysis of the breathing and cortical activity in experimentally-induced apneas or hypopneas [4,5,6]. In support of Younes’s concept, use of hypnotics has been reported to increase the arousal threshold and CO_2_-mediated genioglossus muscle activity [7] and to decrease the apnea hypopnea index (AHI) in OSA patients [8]. Furthermore, pentobarbital sedation has been reported to augment upper airway negative pressure reflex as well as to increase the arousal threshold [9].

Despite physiological evidence suggesting advantages of arousal threshold reduction, cortical arousal was observed to occur at apnea termination even under light sedation [7]. Consequently, Younes’s original hypothesis that cortical arousal is not necessary to restore sleep-disordered breathing has never been critically tested. This hypothesis can only be tested under deep sedation which could prevent cortical arousal. Deep level of sedation is routinely performed to minimize movement of the patients during extended painful endoscopic therapy, such as endoscopic retrograde cholangiopancreatography and endoscopic submucosal dissection (ESD). Recently, we reported the nature of abnormal breathing, such as apneas and hypopneas assessed by polysomnography under deep sedation for ESD [10]. Accordingly, as a secondary analysis of the study, we carefully examined the occurrences of cortical arousal at the termination of apneas and hypopneas in elderly patients under deep propofol sedation for ESD. 

## 2. Materials and Methods

### 2.1. Ethical Approval and Study Subjects

We performed a prospective observational study after obtaining approval from the institutional Ethics Committee (#1902-2014, Graduate School of Medicine, Chiba University, Chiba, Japan) and written informed consent from each subject. Ten adult patients (6 males and 4 females) undergoing ESD surgery for early gastric cancer under propofol sedation participated in the study. Relatively higher incidence of obstructive apneas and hypopneas was found in the elderly patients under propofol sedation as the primary result of the study [10], this secondary explorative analysis of the data was planned specifically focusing occurrence of cortical arousal at the termination of apneas and hypopneas. 

### 2.2. Data Collection

The patient lying on the left side received 2 L/min oxygen through a nasal prong. In order to minimize body movements during the endoscopic submucosal dissection (ESD), relatively deep sedation maintaining Ramsey score 5 to 6 (loss of responses to verbal commands and light tapping on the shoulder, but arousable by painful stimulation) [11] was achieved by continuous intravenous infusion of propofol (1–2 mg/kg/hour) after bolus injection of propofol (1–2 mg/kg) and pentazocine (7.5 mg). Unstable cardiorespiratory abnormalities detected by the routine patient monitoring, including pulse oximeter and electrocardiogram, were treated by changing propofol infusion rate and airway maneuvers as a standard at our institute. 

For research purposes, a standard polysomnography was performed (PSG-1100, Nihon Kohden, Tokyo, Japan). Bilateral central and occipital electroencephalograms (EEG), bilateral electrooculograms (EOG), submental electromyogram (EMG), electrocardiogram, airflow measurement with a nasal pressure prong and an oro-nasal thermistor, thoraco-abdominal wall motions with piezo respiratory effort sensors, oxygen saturation (SaO_2_), and snoring over a microphone were recorded and stored in a computer for later analyses. 

### 2.3. Data Analysis

Polysomnography data were manually reanalyzed for this secondary analysis by a certified sleep technician (RK) and an investigator (SI) using a dedicated computer software (Polysmith, Nihon Kohden, Tokyo, Japan). Consciousness states (awake or sleep) were determined for each 30 s polysomnography recordings by using the criteria of Rechtschaffen and Kales [12], and breathing during sleep was assessed for this study. Sedation efficacy was determined as a ratio between sleep time and sedation period (time from injection of propofol to emergence from the sedation (responses to verbal commands after completion of the ESD procedure)). Apnea and hypopnea were determined by absence of airflow for 10 s or more and more than 50% decrease of the nasal pressure signal for 10 s or more independently of SaO_2_ change, respectively. Apneas and hypopneas were systematically classified by presence or absence of hypoxemia defined as SaO_2_ reduction by 3% or more from the baseline (desaturation, no desaturation), presence or absence of thoraco-abdominal respiratory movements (obstructive, central), and presence or absence of cortical arousal evidenced by the electroencephalograms upon restoration from apneas and hypopneas (cortical arousal, no cortical arousal). Apnea hypopnea index (AHI) was determined as the frequencies of apneas and hypopneas per hour of sleep for each of the category combinations with and without the 3 features of apneas and hypopneas [10].

### 2.4. Statistical Analysis

Arousal rate defined as the ratio between the incidence of respiratory disturbances with cortical arousal and total incidence of the respiratory disturbances occurring in all participants was calculated for each of the categories. Comparison of the arousal rates with and without desaturation was performed by using Chi-square test. Values were presented as frequencies and proportions for categorical data, and means and standard deviations (SD) for continuous variables. A value of *p* < 0.05 was considered statistically significant, and all *p*-values were two sided. All statistical analyses were performed by using a computer software (SigmaPlot 12.0; Systat Software Inc., Point Richmond, CA, USA). 

## 3. Results

### 3.1. Patients’ Characteristics, Details of the Sedation, and Frequency of Respiratory Disturbances

Patients’ characteristics and details of the sedation are presented in Table 1. As summarized in Table 2, polysomnographic recordings were successfully completed in all patients. Despite administration of the sedatives, the sleep state was not always stable and sometimes interrupted by the awake state in association with the surgical procedures resulting in mean sleep efficacy of 81 ± 7% during the sedation period. No slow wave sleep EEG pattern was observed in this study. We identified 245 respiratory disturbances during sleep state in total and the mean total AHI was 15.4 ± 9.2 h^−1^. Thirty-eight more apneas and hypopneas were identified in this secondary analysis than in the primary analysis [10]. Total AHI was significantly associated with total dose of propofol (*r* = 0.673, *p* = 0.033, *n* = 10). AHI without desaturation was significantly greater than AHI with desaturation (*p* = 0.024).

### 3.2. Typical Examples of Apneas and Hypopneas Without Cortical Arousal during Propofol-Induced Sleep

Figure 1 demonstrates typical examples of obstructive apneas with and without cortical arousal. The upper panel of 30 s window (A) evidenced presence of cortical arousal in accordance with abrupt recovery from the first apnea while no sign of cortical arousal was identified during gradual recovery from the second apnea as evidenced by the bottom panel of 30 s window (B). Figure 2 shows an example of occurrences of central and obstructive during propofol-induced sleep state. The airflow gradually increased and no desaturation developed during the first central apnea. No cortical arousal was evident during recovery of central apnea. Desaturation from SaO2 98% to 95% occurred during the long obstructive apnea lasting more than a minute. EEG during recovery from obstructive apnea indicated no cortical arousal while an increase of submental EMG activity was noted. Interestingly, heart rate did not change before and after apneas. Similarly, Figure 3 presents spontaneously-resolved obstructive hypopneas without signs of cortical arousal. Unlike abrupt resolution of obstructive hypopneas during natural sleep, obstructive hypopneas with flattened nasal pressure signal during sedation-induced sleep gradually increased the airflow in accordance with increase of the breathing efforts. Figure 4 demonstrates a series of obstructive apneas and hypopneas. The second obstructive apnea shown by an arrow was resolved by an abrupt increase of airflow which was followed by long-lasting obstructive hypopneas with flattened nasal pressure signal indicating inspiratory flow limitation. No desaturation, no cortical arousal, and no heart rate change were observed throughout the five minutes recording.

### 3.3. Arousal in Response to Apneas and Hypopneas with and without Desaturation

Table 3 presents detailed analysis of occurrence of cortical arousal in response to 245 apneas and hypopneas identified in this study. Cortical arousal only occurred in 55 apneas and hypopneas (22.5%) during propofol-induced sleep. Apneas and hypopneas without arousal and desaturation were most commonly observed (65.7%). Chi-square test indicated that incidence of cortical arousal was significantly associated with occurrence of desaturation and the most common pattern was apneas and hypopneas without arousal and desaturation regardless of the types of the respiratory disturbances. 

## 4. Discussion

This is the first study which assessed the occurrence of cortical arousal in response to apneas and hypopneas during propofol-induced sleep in elderly patients undergoing endoscopic surgery. Cortical arousal only occurred in 55 of 245 apneas and hypopneas (22.5%) during propofol-induced sleep. Apneas and hypopneas without arousal and desaturation was most commonly observed (65.7%) regardless of types of disordered breathing. The results strongly suggest that cortical arousal is possibly a coincidental event to the non-arousal recovery from respiratory disturbances and is not always necessary to restore breathing even in the elderly.

### 4.1. Possible Mechanisms and Consequences of Spontaneous Recovery from Apnea and Hypopnea without Cortical Arousal during Propofol Sedation

Propofol is a potent GABA agonist and used as a general anesthetic as well as a sedative depending on the administration dose [13]. It dose-dependently depresses consciousness level and increases the arousal threshold. It significantly depresses both central and peripheral chemo-sensitivities [14,15]. Propofol concentration depth increase has been reported to significantly reduce the genioglossus activity and increase the critical closing pressure to above the atmospheric pressure in adults [16]. Despite the depression of genioglossus activity and UA negative pressure reflex by propofol injection, increased central and peripheral chemo-stimulation has been reported to restore the reduced genioglossus activity [17]. To our knowledge, this is the first clinical study which assesses the effects of propofol sedation on the consciousness and breathing, and which documents the periodicity of obstructive apnea and hypopnea in the elderly even under sedation.

According to recent accumulated knowledge of arousal and obstructive sleep apnea [6], interactions between the cortical arousal threshold and UA opening and closing thresholds can explain mechanisms of periodic obstructive breathing (Figure 5). When the arousal threshold is below the UA opening threshold (Figure 5A), the cortical arousal occurs before airway opening with an overshoot increase of the respiratory efforts and UA dilating muscle activity leading to unstable large ventilatory oscillation. 

In contrast, propofol increases arousal threshold over the UA opening threshold and decreases respiratory drives to the diaphragm and UA muscles (Figure 5B). It is possible that the increase of UA negative pressure reflex, caused by increase of chemical stimuli during apnea or hypoxemia, may elicit the UA dilating muscle activity to first reach the UA opening threshold, which consequently restores ventilation. Ventilation is maintained until the UA dilating muscle activity decreases below the UA closing threshold which is generally below the UA opening threshold [18]. Due to reduced oscillation of the UA dilating muscle activity, stable breathing can be established despite a reduced UA dilating muscle activity. No cortical arousal is required in this conceptual mechanism of spontaneous recovery from apnea in agreement with the results of this study.

Despite common UA obstruction episodes under propofol sedation, few critical respiratory events occurred in this study, which may reflect few reports of fatal complications under propofol sedation for endoscopic procedures [19,20]. However, we consider that this does not guarantee the safety of propofol sedation and does not deny necessity of close respiratory monitoring during the sedation. As Figure 5C illustrates, deeper propofol sedation would significantly decrease the UA dilating muscle activity below the UA opening threshold resulting in persistent UA closure and the possible development of critical hypoxemia. In fact, higher doses of propofol were associated with increased severity of respiratory disturbances. Without doubt, unnecessarily deeper sedation totally abolishing cortical arousal response to critical respiratory events should be avoided.

### 4.2. Limitations of this Study

There are several limitations in this study. Firstly, the sample size is very small and the patient population is severely limited to non-obese elderly without previous OSA diagnosis. Only few respiratory disturbances occurred during propofol sedation resulting in severe hypoxemia commonly observed in patients with severe OSA. Obviously, our study design does not allow generalization of the findings. However, we believe that incidences and natures of respiratory disturbances and restorative capacity without cortical arousals were objectively assessed by polysomnographies during the sedation. Certainly, four electroencephalograms used in this study were probably not able to cover the whole brain activities and undetected cortical arousal may have occurred during recovery from the respiratory disturbance. However, we consider this possibility unlikely due to the propofol dose used in this study [6,21]. Secondly, we did not directly measure the UA dilating muscle activity and have no evidence for its contribution to the recovery from the respiratory disturbances without cortical arousal. Thirdly, the obstructions and their relief may have been produced by the large endoscope and scope manipulations in the pharynx. In other words, resumption of the flow may not have been due to the spontaneous opening and may have been caused by mechanical pharyngeal opening without cortical arousal. Moreover, the endoscope itself may also be an independent stimulus of dilator activity through stimulating pharyngeal mechano-receptors. Accordingly, future studies should directly assess dynamic changes of the UA dilating muscle activity and ventilatory volume during periodic apneas and hypopneas under sedation without an endoscope within the pharynx. Fourthly, this study was performed in the lateral posture in which gravitational impact on the UA patency is smaller than in the supine posture [22]. Recovery from UA closure may be easier in the lateral position facilitating non-arousal mechanism by decreasing both UA closing and opening thresholds shown in Figure 5. Lastly, this study did not perform polysomnography during natural sleep, preventing comparisons of the severity and patterns of respiratory disturbances between natural sleep and propofol sedation. In particular, the number of participants with obstructive sleep apnea may have increased the rate of respiratory disturbances with severe hypoxemia. However, we consider the nature of recovery from apneas and hypopneas was well characterized in the non-obese elderly patients under deep sedation even without previous OSA diagnosis. It would be an interesting future study to explore the differences of the nature of respiratory disturbances during propofol sedation between patients with and without obstructive sleep apnea [23].

### 4.3. Clinical Implications of the Results of this Study

Effects of various types of hypnotics and sedatives on OSA have been assessed in many previous studies, but the results are diverse; some studies demonstrated an increase of AHI and apnea duration, but clinically meaningful AHI reduction without increase of apnea duration was reported particularly in studies using nonmyorelaxant sedatives such as eszopiclone [8,24]. Jordan et al. suggested selected patients with phenotypes such as lower UA collapsibility, better UA dilator muscle function, lower arousal threshold, and high ventilator response to arousal as a candidate for successful OSA treatment with hypnotics [24]. Our results agree with their concept and support possible OSA treatment with hypnotics.

Pharmacological sedation is becoming frequently used in various clinical situations such as invasive endoscopy, minor surgery, radiological examinations for children, and mechanical ventilation in ICU [10,25,26,27]. Apneas and hypopneas during sedation were spontaneously terminated either with or without cortical arousal in our patients. However, these could be indicators for early effective prevention of the critical events during and/or immediately after sedation. Although more evidence is necessary, unexpectedly-deeper sedation which could impair cortical arousal respiratory compensatory mechanisms, as illustrated in Figure 5C, which may be involved in the pathogenesis of the rare, but critical cardio-respiratory complications require intensive interventions or treatments during propofol sedation for GI endoscopy [28]. More severe hypoxemia could develop when oxygen therapy is ceased immediately after endoscopy when residual sedatives could worsen respiratory disturbances. In fact, deaths in patients undergoing GI endoscopy during and after propofol sedation have been reported [19]. Clearly, future studies need to explore clinical significance of the non-hypoxemic respiratory disturbances and roles of the respiratory monitoring such as capnography, nasal pressure, snoring sound measurement during sedation [25,29,30,31]. 

## 5. Conclusions

Obstructive apneas and hypopneas during propofol sedation were spontaneously terminated without cortical arousal while repeated periodically. The appropriative depth of propofol sedation appears to reduce both the UA dilating muscle activity and its oscillation, possibly stabilizing respiration and decreasing frequency of respiratory disturbances. However, extensive suppression of the arousal threshold with higher doses of propofol can lead to prolonged apnea and critical hypoxemia suggesting careful respiratory monitoring with both pulse oximetry and nasal pressure monitor during advanced GI endoscopy under propofol sedation.

## Figures and Tables

**Figure 1 ijerph-16-03482-f001:**
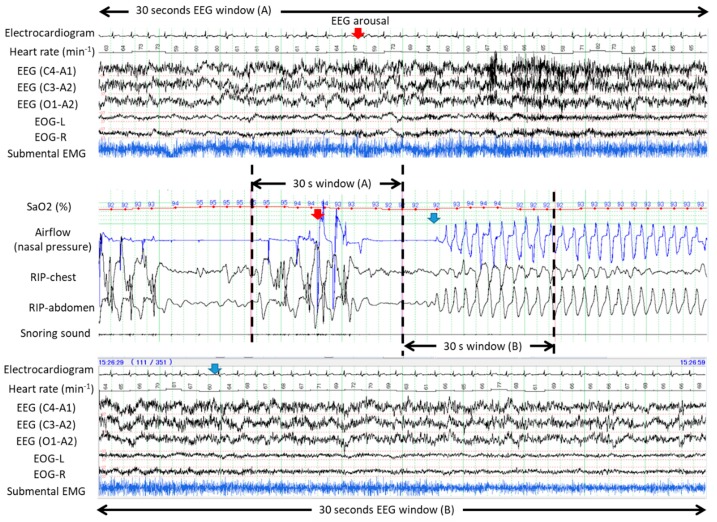
Five-minute polysomnographic tracings demonstrating restorations from obstructive apneas with and without cortical arousal during endoscopic submucosal dissection (ESD) procedure under propofol sedation. The upper panel of 30 s window (**A**) evidenced presence of cortical arousal in accordance with abrupt recovery from the first apnea while no cortical arousal occurred during gradual recovery from the second apnea as evidenced by the bottom panel of 30 s window (**B**).

**Figure 2 ijerph-16-03482-f002:**
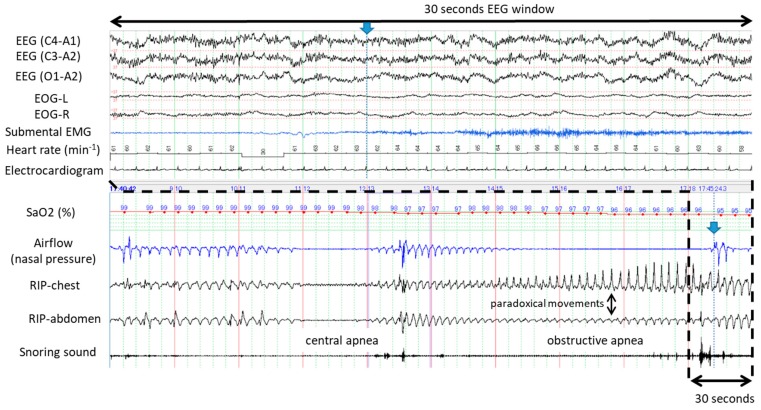
Five-minute polysomnographic tracings demonstrating restorations from long central and obstructive apneas without desaturation and cortical arousal during ESD procedure under propofol sedation. The arrow indicates abrupt termination of obstructive apnea without electroencephalograms (EEG) change suggesting cortical arousal.

**Figure 3 ijerph-16-03482-f003:**
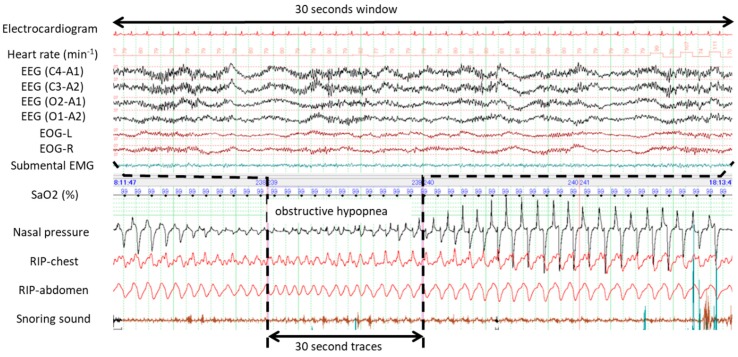
Typical polysomnographic example of obstructive hypopnea occurred during ESD procedure under propofol sedation. No cortical arousal was detected by the electroencephalograms (EEG) during the recovery from hypopnea. Note inspiratory flow limitation pattern of the nasal pressure signal and progressive increase of the inspiratory period. Also note the gradual increase of the respiratory flow during recovery from hypopnea unlike the sudden increase of the respiratory flow commonly seen in obstructive hypopnea during natural sleep.

**Figure 4 ijerph-16-03482-f004:**
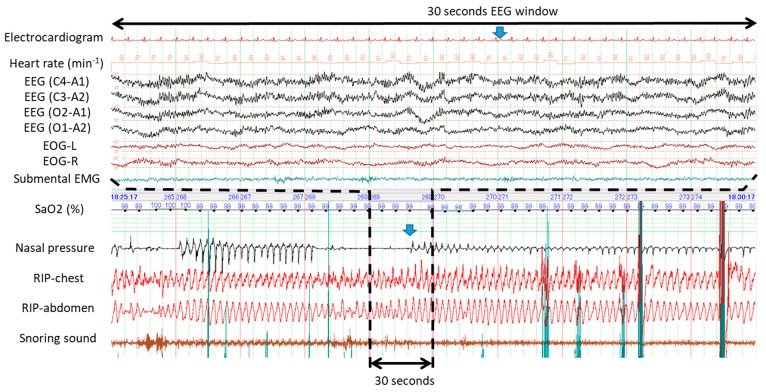
Five minute polysomnographic recordings of a series of obstructive apneas and hypopneas during propofol-induced sleep. Note that the second obstructive apnea shown by the arrow was resolved by an abrupt increase of airflow which was followed by long-lasting obstructive hypopneas with flattened nasal pressure signal indicating inspiratory flow limitation. No desaturation, no cortical arousal, and no heart rate change were observed throughout the 5-minute recording.

**Figure 5 ijerph-16-03482-f005:**
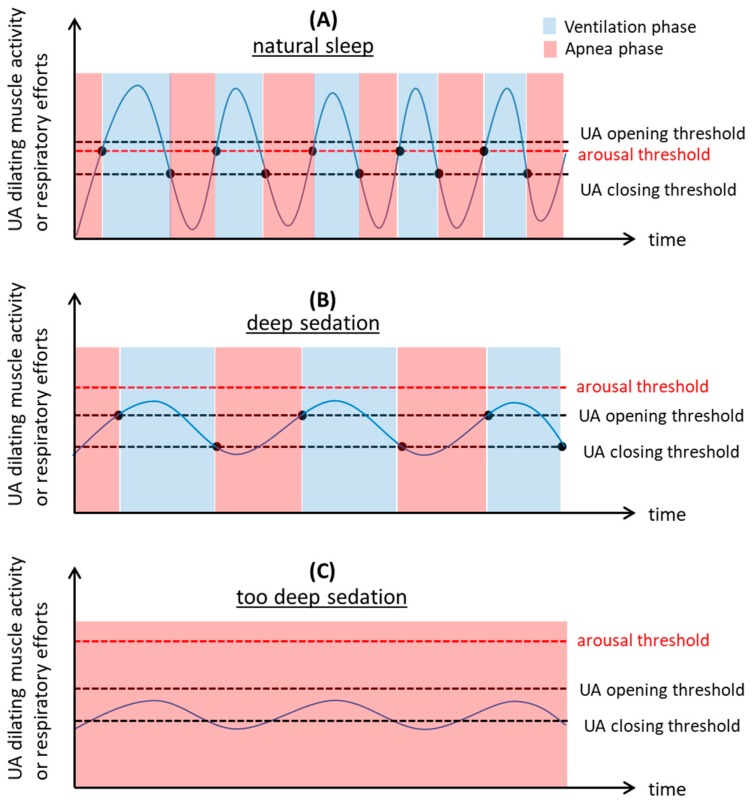
Interactions between the cortical arousal threshold and upper airway (UA) opening and closing thresholds for explaining the mechanisms of periodic obstructive breathing. Sinusoidal curves represent changes of UA dilating muscle activity and respiratory efforts. Intersections (closed circles) between the curves and threshold lines correspond to the timings for the beginning and end of obstructive apneas. (**A**) Interactions commonly seen during natural sleep. When the arousal threshold is below the UA opening threshold, the cortical arousal occurs before airway opening with overshoot increase of the respiratory efforts and UA dilating muscle activity leading to an unstable large ventilatory oscillation. (**B**) Possible interactions during deep sedation. When a sedative increases arousal threshold over the UA opening threshold and decreases respiratory drives to the diaphragm and UA muscles, the UA dilating muscle activity first reaches UA opening threshold restoring ventilation without cortical arousal. The ventilation is maintained until the UA dilating muscle activity decreases below the UA closing threshold which is generally below the UA opening threshold. (**C**) Possible interactions when the sedation is accidentally too deep. Because of profound reduction of the UA dilating muscle activity below the UA opening threshold, UA closure persists possibly leading to prolonged apnea and critical hypoxemia.

**Table 1 ijerph-16-03482-t001:** Patients’ characteristics and propofol doses for the sedation.

Variables	*n* = 10
	Mean ± SD or N
Demographics	
Age (years)	71 ± 7
Males, Females	6, 4
Height (cm)	159 ± 9
Body weight (kg)	59 ± 8
Body mass index (kg/m^2^)	23.6 ± 3.5
Sedation drug	
Initial injection dose of propofol (mg/kg)	1.2 ± 0.4
Total dose of propofol (mg/kg/hour)	5.1 ± 0.8

SD: standard deviation, N: number of patients.

**Table 2 ijerph-16-03482-t002:** Details of the sedation and nature of respiratory disturbances.

Variables	*n* = 10
	Mean ± SD
Sedation quality	
Sedation period (min)	114 ± 36
Total sleep time (min)	93 ± 34
Sedation efficacy (%)	81 ± 7
Sedation efficacy (%)	15.4 ± 9.2
Nature of respiratory disturbances during sedation	
Apnea hypopnea index with desaturation (h^−1^)	4.0 ± 5.3
Apnea hypopnea index without desaturation (h^−1^)	11.4 ± 7.1
Mean duration of apnea hypopnea (seconds)	26 ± 2
Longest apnea and hypopnea (seconds)	62 ± 18
Mean nadir SaO_2_ of desaturation events (%)	89.6 ± 5.1
Lowest SaO_2_ of desaturation events (%)	82.0 ± 11.7

SD: standard deviation

**Table 3 ijerph-16-03482-t003:** Results of analysis of occurrence of cortical arousal in response to 245 apneas and hypopneas. Values are frequencies (proportion).

Types of Respiratory Disturbances	Arousal	Desaturation Events: N (%)	No Desaturation Events: N (%)	Total Events N (%)	Chi-Sqaure *p* Value
Apnea and hypopneas (*n* = 245)	yes	24 (9.8)	31 (12.7)	55 (22.4)	<0.001
	no	29 (11.8)	161 (65.7)	190 (77.6)	
Obstructive apnea (*n* = 134)	yes	16 (11.9)	16 (11.9)	32 (23.8)	<0.001
	no	19 (14.2)	83 (61.9)	102 (76.2)	
Obstructive hypopnea (*n* = 61)	yes	8 (13.1)	5 (8.2)	13 (21.3)	0.012
	no	10 (16.4)	38 (62.3)	48 (78.7)	
Central apnea (*n* = 19)	yes	0	1 (5.3)	1 (5.3)	NA
	no	0	18 (94.7)	18 (94.7)	
Central hypopnea (*n* = 31)	yes	0	9 (29.0)	9 (29.0)	NA
	no	0	22 (71.0)	22 (71.0)	

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
