# Peer review of "Occurrence of Cortical Arousal at Recovery from Respiratory Disturbances during Deep Propofol Sedation"

_ijerph, 2019, doi:10.3390/ijerph16183482_

Round 1
Reviewer 1 Report
General comments
This manuscript examines the presence/absence of arousal following respiratory disturbance under deep propofol sedation. The purpose of the manuscript is to characterise whether the upper airway can reopen without arousal. It is a simple straightforward paper that demonstrates that a majority of the time, under propofol sedation airway patency can be restored without presence of a cortical arousal on the EEG. The paper provides support for the theory proposed by Magdy Younes that arousal is a coincidental event at apnea/hypopnea termination and that in many instances arousal may be unnecessary. The manuscript is well presented and clear. However, the study is quite limited in that it does not examine the mechanisms of restoration of airway patency, the sample is elderly non-obese patients without sleep apnoea and a majority of events were not associated with desaturation, therefore the generalisability is questionable. However, the authors provide good coverage of the key limitations and next steps. While the study does not offer a substantial advancement in understanding OSA pathogenesis it does offer some novel findings that may assist in future work.
Specific comments:
Introduction
Hypnotic studies do not always show an improvement in AHI, this is even when an increase in the arousal threshold is observed. I think it is worth presenting a more balanced opinion on this. There is a review on this topic that is worth considering Jordan 2017 – Physiology of arousal in obstructive sleep apnea and potential impacts for sedative treatment.Methods
Patients were studied on the left side. There are well known effects of position of sleep apnea, would the results have been different if they were assessed supine. E.g. due to greater gravitational effects and therefore load on the upper airway would they have required greater incidence of arousal to reopen their airway Were these patients previously diagnosed with apnea? You state apneas and hypopneas were scored as a 50% reduction in nasal pressure but how did you differentiate the two?Results
Can sleep stage be under propofol sedation? If so, did you have enough events to see if there was a difference by sleep stage e.g. N2 vs. SWS.? Table 1 and 2, require headings above the variables. E.g. Table 1 “demographics and sedation characteristics” should be on top of the first column and “mean ± SD” should be on top of the second column. Worth including a figure with an arousal so that readers can see the difference in EEG.Discussion
Your patients were very light with a mean BMI of 23.6. Do you think that these results can be generalised to OSA patients who are generally obese or greater? Would the non-arousal mechanisms struggle to open the airway more in this patient type? I think this is worth discussing. There was a low incidence of events associated with desaturations (e.g. AHI without desaturation 4 per hour). Therefore, a majority of the event types were likely more “mild” in that they did not cause chemical disturbance. Again do you think you can generalise from your sampled events to more severe events that do results in respiratory imbalance? You mention a few critical events occurred. What were they? Is it worth describing? In the clinical relevance section, it may be beneficial to have something on this work supporting the use of sedatives such as Zopiclone to treat OSA.Minor language corrections:
In the title it should read “at recovery from…”. In the introduction perhaps replace “small children” with “young children”. In the introduction change from “Despite these physiological evidences” to “Despite physiological evidence”. You state “Primary results of the study were previously reported (10) and this is an explosive secondary analysis…” I don’t think explosive is the right word. Did you mean explorative?Author Response
Please see the attachment.

Reviewer 2 Report
Reviewer’s comments and suggestions for Authors
In this manuscript, the authors examined the incidences of cortical arousal at the termination of apneas and hypopneas in elderly patients who received propofol sedation that increases the cortical arousal threshold point. The study used (n=10) elderly patients undergoing advanced endoscopic procedures under propofol-sedation.
The study result reported that cortical arousal only occurred in 55 apneas and hypopneas (22.5%), and apneas and hypopneas without arousal and desaturation were most commonly observed (65.7%) that was not dependent on the types of disordered breathing.
In conclusion, they have reported that even under deep propofol sedation, apneas and hypopneas can be terminated without cortical arousal. However, extensive repression of the arousal threshold that leads to critical hypoxemia.
The manuscript is well written in terms of the English language. The topic is of great interest for clinicians and basic researchers working in sleep research. However, some of the things need to be considered in the manuscript by reviewing a few minor concerns.
(1) In the abstract part, the author has to discuss completely the following sentence in a better way “Chi-square test indicated that incidence of cortical arousal was significantly associated with occurrence of desaturation.
(2) The author needs to confirm why they need a secondary analysis of the study, or they have to make a hypothesis for this study. Need to mention some references based on parameter taken in the study.
(3) Line 66- 67, the word explosive should be changed with some other word.
(4) Line 97-98, Please elaborate more on this index “Apnea hypopnea index (AHI) was determined as the frequencies of apneas and hypopneas per hour of sleep regardless of the desaturation for each of the categories”.
(5) Page number 103, the Summary statistics word need to modify.
(6) The first letter of all the given table needs to be capitalized.
(7) Line number 208, the spelling of doses was wrong
(8) The study needs to explain how they defend the limitation that they included in the manuscript. “this study did not perform polysomnography during natural sleep, preventing comparisons of the severity and patterns of respiratory disturbances between natural sleep and propofol sedation”
(9) The conclusion first line need to make simpler “Obstructive apneas and hypopneas during propofol sedation were spontaneously terminated without cortical arousal but repeated alike OSA during natural sleep”.
(10). Please check the references 23,24,25,29. The style was not based on the journal guidelines.
